# Influence of drone carriage material on maintenance of storage temperature and quality of blood samples during transportation in an equatorial climate

**Mohamed Afiq Hidayat Zailani**[1], **Raja Zahratul Azma Raja Sabudin**[1], **Aniza Ismail**[2], **Rahana Abd Rahman**[3], **Ismail Mohd Saiboon**[4], **Shahnaz Irwani Sabri**[5], **Chan Kok Seong**[6], **Jamaludin Mail**[6], **Shamsuriani Md Jamal**[4], **Gan Kok Beng**[7], **Zaleha Abdullah Mahdy**[3]*

1 Department of Pathology, Faculty of Medicine, Universiti Kebangsaan Malaysia (UKM), Kuala Lumpur, Malaysia, 2 Department of Community Medicine, Faculty of Medicine, Universiti Kebangsaan Malaysia (UKM), Kuala Lumpur, Malaysia, 3 Department of Obstetrics and Gynecology, Faculty of Medicine, Universiti Kebangsaan Malaysia (UKM), Kuala Lumpur, Malaysia, 4 Department of Emergency Medicine, Faculty of Medicine, Universiti Kebangsaan Malaysia (UKM), Kuala Lumpur, Malaysia, 5 Queen Elizabeth II Hospital (QEHII), Ministry of Health (MOH), Kota Kinabalu, Malaysia, 6 Sabah Women and Children Hospital (SWACH), Ministry of Health (MOH), Kota Kinabalu, Malaysia, 7 Department of Electrical, Electronic & Systems Engineering, Faculty of Engineering and Built Environment, Universiti Kebangsaan Malaysia, Kuala Lumpur, Malaysia

* zaleha@ppukm.ukm.edu.my

**Data Availability Statement:** All relevant data are within the paper and its Supporting Information files.

## Abstract

The disruptive potentials of drones are rapidly growing including for the delivery of blood samples in healthcare. Maintenance of the quality of blood samples is important to ascertain that the drone is a safe mode of transportation, particularly during emergencies and in critical cases. The influence of the drone carriage material on blood samples transportation was investigated in this study. Two phases of drone simulation flights were conducted in Cyberjaya, Malaysia. In Phase 1, the effect of drone carriage material on the internal storage temperature during blood samples transportation was determined. Three types of carriage materials were compared: aluminium, expanded polystyrene (EPS) foam, and polypropylene (PP) plastic. In Phase 2, the quality of drone-transported blood samples was assessed, using the best material from Phase 1 as the drone carriage material. Biochemical and hematological analyses of 60 blood samples were conducted using five parameters. In Phase 1, EPS foam was found to be the best material to maintain a stable and favorable internal storage temperature at mean kinetic temperature ±SD of 4.70 ±1.14˚C. Much higher and unfavorable mean kinetic temperatures were recorded for aluminium (11.46 ±0.35˚C) and plastic (14.17 ±0.05˚C). In Phase 2, laboratory tests show that the quality of blood samples was well maintained, and the mean biochemical and hematological parameters of drone-transported blood samples showed no significant alteration compared to ground controls. Drone carriage material is an important determinant of the quality of blood samples transported by drone, particularly in hot equatorial climates as in Malaysia. The blood storage temperature was best maintained using EPS foam, as evidenced by the favorable average temperature and preservation of hematological and biochemical parameters of the blood samples.

**Funding:** This research was funded by UNIVERSITI KEBANGSAAN MALAYSIA (https://www.ukm.my/portal/), grant number DCP-2018–004/1 and was received by ZAM. The funder had no role in study design, data collection and analysis, decision to publish, or preparation of the manuscript.

**Competing interests:** The authors have declared that no competing interests exist.

## Introduction

Disruptive technology is an innovation that significantly changes the conventional operational manner of industries, businesses, and systems, such as drones or Unmanned Aerial Vehicles [1, 2]. The high versatility of drones leads to its variety of applications including in the fields of healthcare, agriculture, surveillance, commerce, military, and urban development [3–9]. Drones have been proven to increase geographical accessibility [10], reduce transportation time [11], lower carbon footprint [12], and improve cost-effectiveness of delivery services [13, 14].

In the healthcare sector, drones were explored as one of the potential alternative solutions in blood samples transportation [15]. While a number of laboratory tests requiring high-tech equipment are only available at tertiary health facilities, many hospitals and clinics in developing countries such as Malaysia are scattered in resource-limited remote locations in order to cater for the widely distributed population in the country [16]. As an example, district hospitals and rural clinics in the states of Sabah and Sarawak (East Malaysia) are situated around 35 to 60 minutes' drive away from the nearest tertiary hospital. Hence, efficient transportation is essential to deliver blood samples for relatively complex yet frequently requested tests such as full blood picture (FBP), thyroid function test (TFT), and iron studies.

Other than distance, blood samples transportation by drone faces another critical geological factor–the equatorial climate. The Malaysian weather is hot, humid, and rainy throughout the year. Annually, average temperature across Malaysia is between 23°C and 34°C with average 80% annual rainfall (between 2000mm to 2500mm) and 80.5% annual percentage of humidity [17]. This environmental factor poses a challenge in selecting the best carriage material in order to keep blood samples within an optimal storage temperature throughout transportation, to prevent jeopardizing the quality of the blood samples. The current practice of logistical transportation of blood samples in Malaysia uses ground transportation such as motorcycles and ambulances. Once blood samples have been collected into tubes, they are stored inside a cooler box made of expanded polystyrene (EPS) foam material, with ice packs to maintain the internal storage temperature, and subsequently delivered to the nearest tertiary hospital.

For example, in the diagnosis of Glucose-6-Phosphate Dehydrogenase (G6PD) deficiency, the enzymatic level of blood samples that were collected in Ethylene Diamine Tetra-acetic Acid (EDTA) tubes which were found to be more stable and the diagnostic results more accurate when the sample is stored in low temperatures of 2 to 8°C [18]. Ashworth M et al. (2021) reported that whole-blood samples stored at 4°C has been shown to manifest fewer changes in plasma cytokine levels compared to samples held at room temperature [19]. Failure to deliver these blood samples within the correct temperature requirement over a long distance may thus alter the biochemical properties of the blood, affect the diagnosis and clinical management, and harm patients particularly vulnerable groups such as pregnant women, cancer patients, premature babies, and thalassemia patients.

The main objective of our study was to determine the most suitable material to be used as the drone carriage in order to maintain a favourable blood storage temperature within an equatorial climate, and its impact on the quality of blood samples carried by the drone. Results of the hematological and biochemical analysis of the flown blood samples would serve as evidence of the drone carriage material's suitability to transport blood samples by drone in a tropical weather.

## Materials and methods

### Study design

A comparative experimental analysis was conducted to identify the effect of drone carriage materials on blood samples transportation. Drone flight simulations were carried out on two

**Table 1. Characteristics of the materials used for drone flights F1, F2 and F3.**

| Type of material | Aluminium | Polypropylene (PP) plastic | Expanded polystyrene (EPS) foam |
|---|---|---|---|
| Chemical composition | Silvery white metal element of Group 13 of the periodic table. | A transparent thermoplastic made from the combination of propylene monomers. | White foam plastic material produced from solid beads of polystyrene (hydrocarbon compound). |
| Density (kg/m$^3$) | 2680–2700 | 910 | 12–46 |
| Elastic modulus (Gpa) | 70–80 | 1.1–1.6 | 0.00650–2.65 |
| Tensile strength (Mpa) | 124–290 | 27 | 0.8–1.1 |
| Yield strength (Mpa) | 195 | 35–40 | 47–51 |
| Thermal conductivity [W/(m·K)] | 151–202 | 0.1–0.2 | 0.035–0.037 |
| Elongation at break (%) | 18–33 | 50–145 | 5–13 |
| Corrosion rate (μm/year) | 0.8–0.28 | < 0.1 | < 0.1 |
| Cost (USD/kg) | 0.93 | 1.42 | 0.98 |
| Environmental impact | • Energy intensive (water, electricity, and resource) | • Slow decomposition (20–30 years) | • Easily recyclable |
| | | | • Non-biodegradable |
| | • Greenhouse gas emissions | • Toxic additive | |

different days as Phase 1 and Phase 2 of the experiment. Phase 1 aimed to identify the most suitable material for drone carriage to ensure that the internal storage temperature was maintained at the optimal favourable level. Three types of carriage materials were used: aluminium (Grade 6061, 1 mm, Smartiff Ptd. Ltd., Malaysia), polypropylene (PP) plastic (Grade MMBX-8501, 3mm, Mysuppliersorg Pte. Ltd., Malaysia), and expanded polystyrene (EPS) foam (Grade NR-4153, 10mm, Mr. DIY Pte. Ltd., Malaysia), for flights F1, F2 and F3 respectively.

The selected materials were chosen based on their capability to be made into a lightweight storage with good thermal insulation that is important to ensure safe drone transportation of blood samples. The physical characteristics of each material are summarized in Table 1 [20–24]. These drone carriages were within the optimum specification requirements for a reasonably sized and economically viable drone with a payload of 2.5 kilograms, that is planned for such a function in the long term. By excluding the weight of the contents inside the carriage such as blood samples, icepacks and thermologger totalling 1.5 kilograms, the ideal weight of the carriage therefore should not exceed 1 kilogram. In this study, the actual weights of the three carriages used were 0.2, 0.6 and 0.9 kilogram for PP plastic, EPS foam and aluminium, respectively. Their internal storage volume ranged from 4 to 6 liters, and their external dimensions (length x width x height) were 29 cm x 20 cm x 17 cm for PP plastic, 24 cm x 14 cm x 18 cm for EPS foam and 30 cm x 30 cm x 8 cm for aluminium.

Once the best drone carriage material has been determined, we embarked on Phase 2 of the simulation (drone flights F4 and F5) using the best material from Phase 1. Phase 2 aimed to investigate the quality of drone-transported blood samples using the selected drone carriage material. Six tubes of blood samples were collected on the morning of the event from each of ten verbally consented healthy donors through convenience sampling (60 tubes of blood samples in total). The number of samples was determined based on the weight of the samples versus the payload capacity of the drone. For this study, the drone model used had a payload capacity of 6.8 kilograms. After considering multiple external factors affecting the payload capacity and performance of the drone such as battery weight, drone weight, propeller number and size, and flight distance, we limited the total amount and weight of collected blood samples with their packaging and icepacks to suit our long term target of using a drone with a payload capacity of 2.5 kilograms [25].

All blood samples were drawn using a standard phlebotomy technique with a volume of 20 mL of blood from each subject. The blood samples were collected in three labelled EDTA tubes (PUTH® Vacumine, Chengdu, China. Batch No. 2002092A) and three labelled plain tubes (PUTH® Vacumine, Chengdu, China. Batch No. 2005096C). Subsequently, the blood samples were divided into three batches according to the labels F4, F5 and controls. As a safety precaution, blood samples for the drone flights F4 and F5 were packed according to the UN3373 medical packaging regulations (Biological substance, Category B) [26]. This includes three compulsory components, namely the primary receptacles (the EDTA blood tubes) that were encapsulated and protected inside a secondary packaging of Low Density Polyethylene (LDPE) biohazard specimen bags, and lastly an outer packaging for transport (the drone carriage). The controls referred to blood samples that remained on the ground to imitate ground transportation.

Once both simulation flights F4 and F5 have been conducted, all blood samples (test and control) were immediately brought to the Haematology and Chemical Pathology Units, Department of Diagnostic Laboratory Services, Universiti Kebangsaan Malaysia Medical Centre (UKMMC) for hematological and biochemical analyses. The interval between blood sampling and processing at the laboratory was 3 hours.

Five parameters were selected for laboratory analysis in this study: hemoglobin (Hb) level, hematocrit (Hct), hemolysis index (IH), sodium (Na), and potassium (K) levels. These parameters were chosen based on standard quality control principles for blood samples analysis practiced in our pathology laboratory. Full blood count (FBC) tests were run using a Sysmex XN 3000 Hematology Analyzer (Sysmex Asia Pacific Pte Ltd, Singapore). Serum electrolyte and hemolysis index (BUSE-HIL) tests were performed using an Abbott Architect c16000 Clinical Chemistry Analyzer (Abbott Laboratories (M) Pte Ltd, Malaysia).

## Ethical approval

This study was registered under the National Medical Research Register (NMRR) of Malaysia. Ethical approval was obtained from the Medical Research and Ethics Committee (MREC), Ministry of Health, Malaysia (Reference number: NMRR-19-1801-45727 IIR), and the Universiti Kebangsaan Malaysia (UKM) Research Ethics Committee (Reference number: UKM PPI.800-1/1/5/JEP.2019.420).

## Patient and public involvement

Blood samples were taken from ten healthy donors following verbal consent. Information on the research were given to donors prior to obtaining their consent. It included the purpose and brief methodology of the research, the blood parameters that were to be measured, and the liberty of the donors to withdraw from the study at any time.

## Flight protocol

Flight simulations were carried out using a MATRICE 600 Pro DJI DAT 1.17 multi-rotor drone model that was flown by a certified professional drone pilot from the Aerodyne Group Pte Ltd (Fig 1) [27]. The drone was flown within Visual Line of Sight (VLOS) with an average velocity of 42.9 km/h at an altitude of 300 feet above ground in a designated drone flying zone in Cyberjaya, Malaysia. The drone flight path was set for an 8.15 km flight distance, which was optimal for the drone's battery consumption and failsafe settings (Fig 2). F1, F2 and F3 flights of Phase 1 were conducted consecutively on the same morning, starting at 10:20 a.m. for F1, 11:10 a.m. for F2, and 11:40 a.m. for F3. Phase 2 flights (F4 and F5) were conducted on another morning, starting at 9:05 a.m. for F4 and 9:25 a. m. for F5.

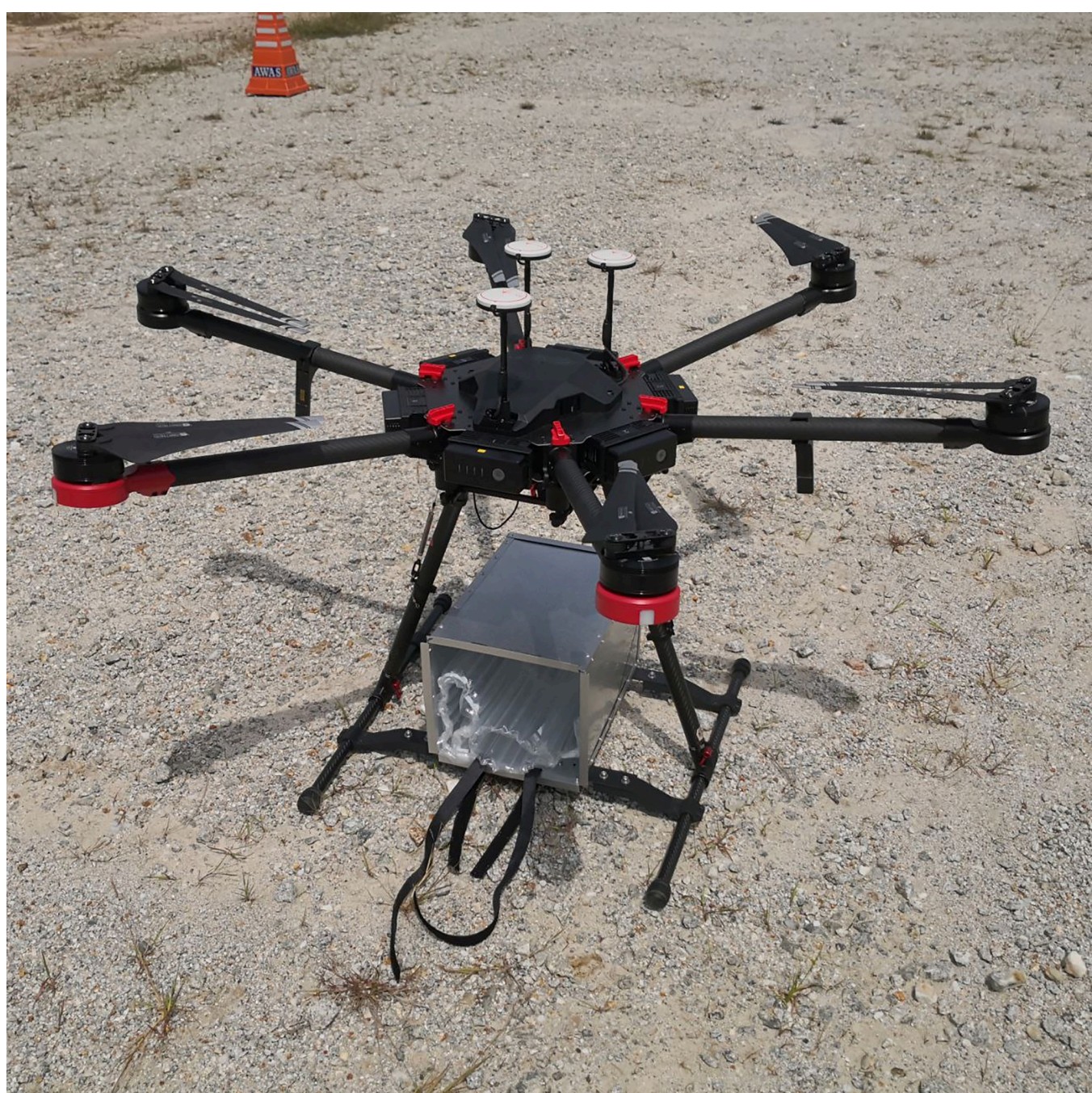

**Fig 1. Multi-rotor drone.** The multi-rotor drone that was used to transport our blood samples.

For Phase 1 flights (F1, F2 and F3), the carriage was filled with 2 units of simulated blood, ice packs, and a calibrated Fourtec MicroLite USB Datalogger LITE5032P-RH model (Fourtec Technologies Pte Ltd, Kuala Lumpur, Malaysia) with a total payload of 1.55 kg. For Phase 2 flights (F4 and F5), similar items were carried along with the blood samples in order to achieve the same total payload. We used simulated blood units consisting of distilled water that was dyed red (Star Brand Cochineal Red Artificial Food Colouring, Malaysia) instead of real blood

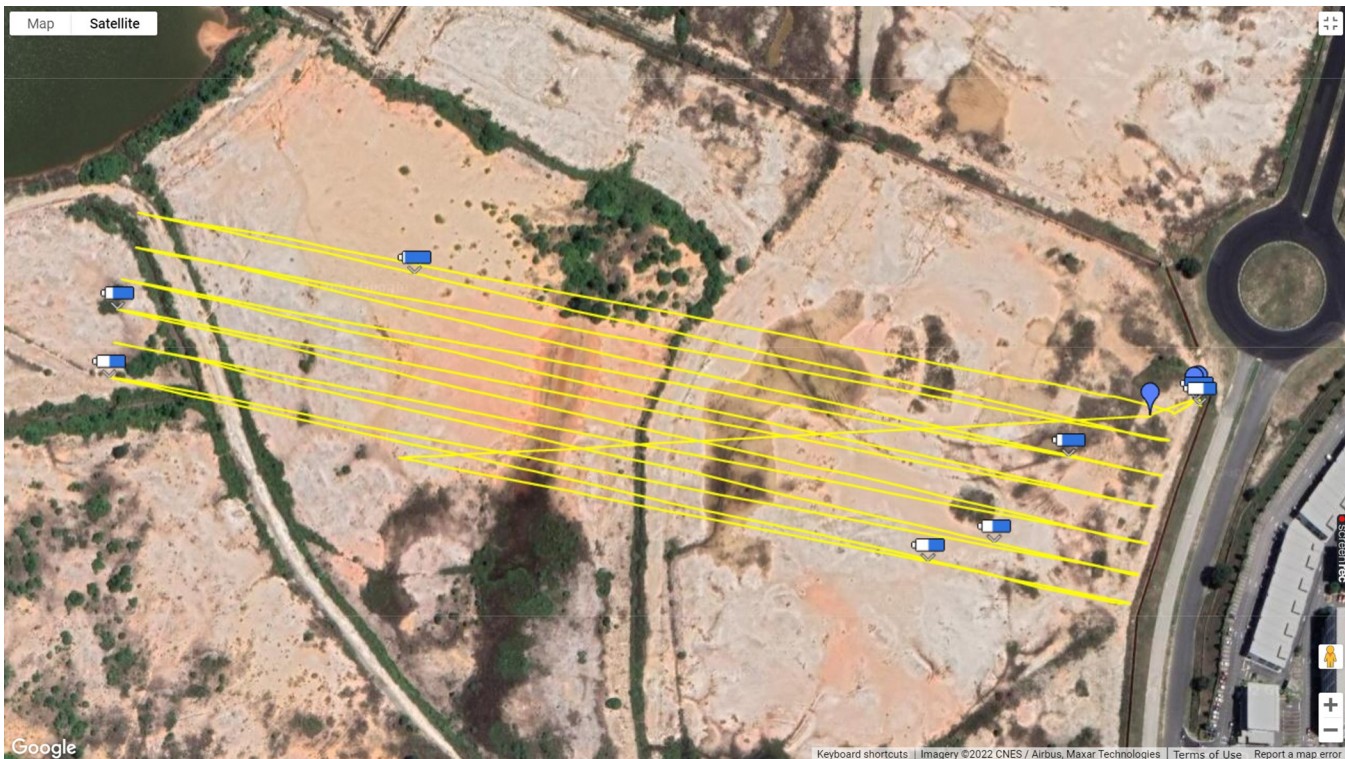

**Fig 2. Drone flight path.** Yellow lines illustrate the drone flight path used for all flights in this research.

packs in view of the highly valuable nature of real blood products and to eliminate the risk of a biohazard mishap in case of a drone crash or fall. The payload was chosen according to the maximum capacity of our drone model. The internal storage temperature of the drone carriage was continuously captured by the datalogger and subsequently analyzed using a Fourtec Data-Suite Version 2.5.4. 12A software. Fig 3 shows the attachment of the carriages to the drone.

## Statistical analysis

Mann-Whitney U test was used to compare the laboratory results of the test blood samples against the controls for hemoglobin level (Hb), hematocrit (Hct), hemolysis index (IH), sodium (Na), and potassium (K) levels. Any difference was considered statistically significant if $P > 0.05$.

## Reporting

The report of this research was made in accordance with the principles of the Transparent Reporting of Evaluations with Nonrandomized Designs (TREND) statement checklist (S1 File) [28]. This is to ensure transparent reporting in an intervention study and facilitate the synthesis of the findings.

## Results

Data collection took place on August 13, 2020 (Phase 1) and November 19, 2020 (Phase 2) in Cyberjaya, Malaysia. The environmental ambience on the morning of both days were approximately similar, with sunny weather and an average ambient temperature of 24.5°C, air

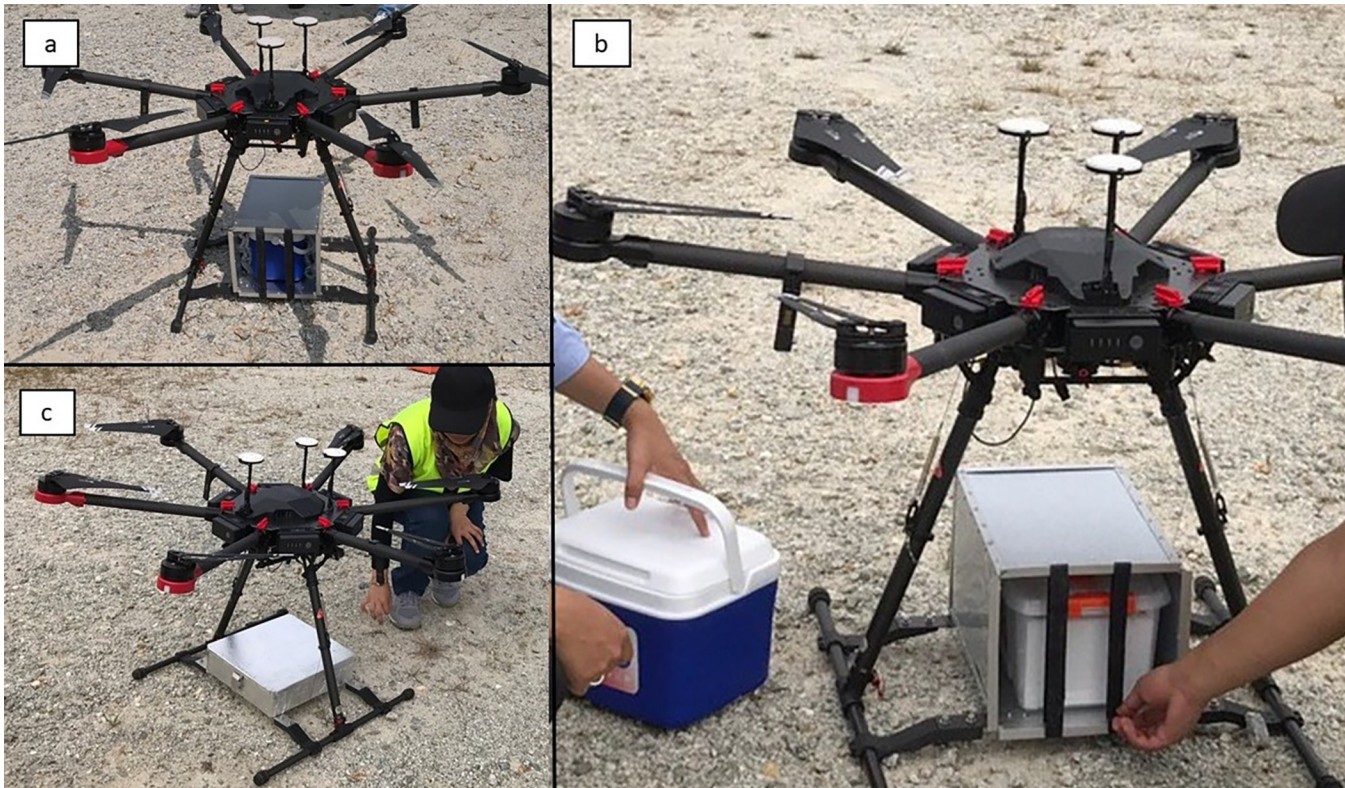

**Fig 3. Attachment of carriages to drone.** (a) Drone carriage made from EPS foam was attached to drone using a custom-made steel box with air cushion bags. (b) Drone carriage made from PP plastic was attached to drone using similar method as EPS foam. (c) Drone carriage made from aluminium was attached directly to drone using customized steel brackets.

humidity of 78%, and wind speed of 5 km/h. During the 17-minute flight, the mean kinetic temperature ± SD of the aluminium carriage of F1 recorded by the datalogger was 11.46 ±0.35°C, whereas in the plastic carriage of F2 the recorded temperature was 14.17 ±0.05°C. In contrast, the EPS foam carriage of F3 showed the lowest average temperature with the most stable result compared to the aluminium and plastic, being able to maintain an average internal storage temperature of 4.70 ± 1.14°C which is optimal for blood samples delivery (Fig 4). Based on these results, the EPS foam carriage was used in Phase 2 (flights F4 and F5) of the second drone simulation.

Blood sample parameters from Phase 2 showed no significant variations in haematological and biochemical parameters compared to controls (Tables 2 and 3).

## Discussion

This study investigated the effect of drone carriage material on storage temperature and blood sample quality during transportation by drone in an equatorial climate using five most common hematological and biochemical laboratory tests, which were Hb, Hct, IH, Na, and K. It is crucial to ascertain which material is best able to maintain the internal storage temperature throughout transport so as to prevent deterioration of blood sample parameters that can jeopardize the accuracy of blood sample results [29]. Our observation showed that EPS foam shaped into a six-faced cuboid with external dimensions (length x width x height) of 24 cm x 14 cm x 18 cm was the best material to be used for drone carriage due to its ability to maintain optimal mean kinetic temperature for blood sample transportation, which was 4.70 ± 1.14°C.

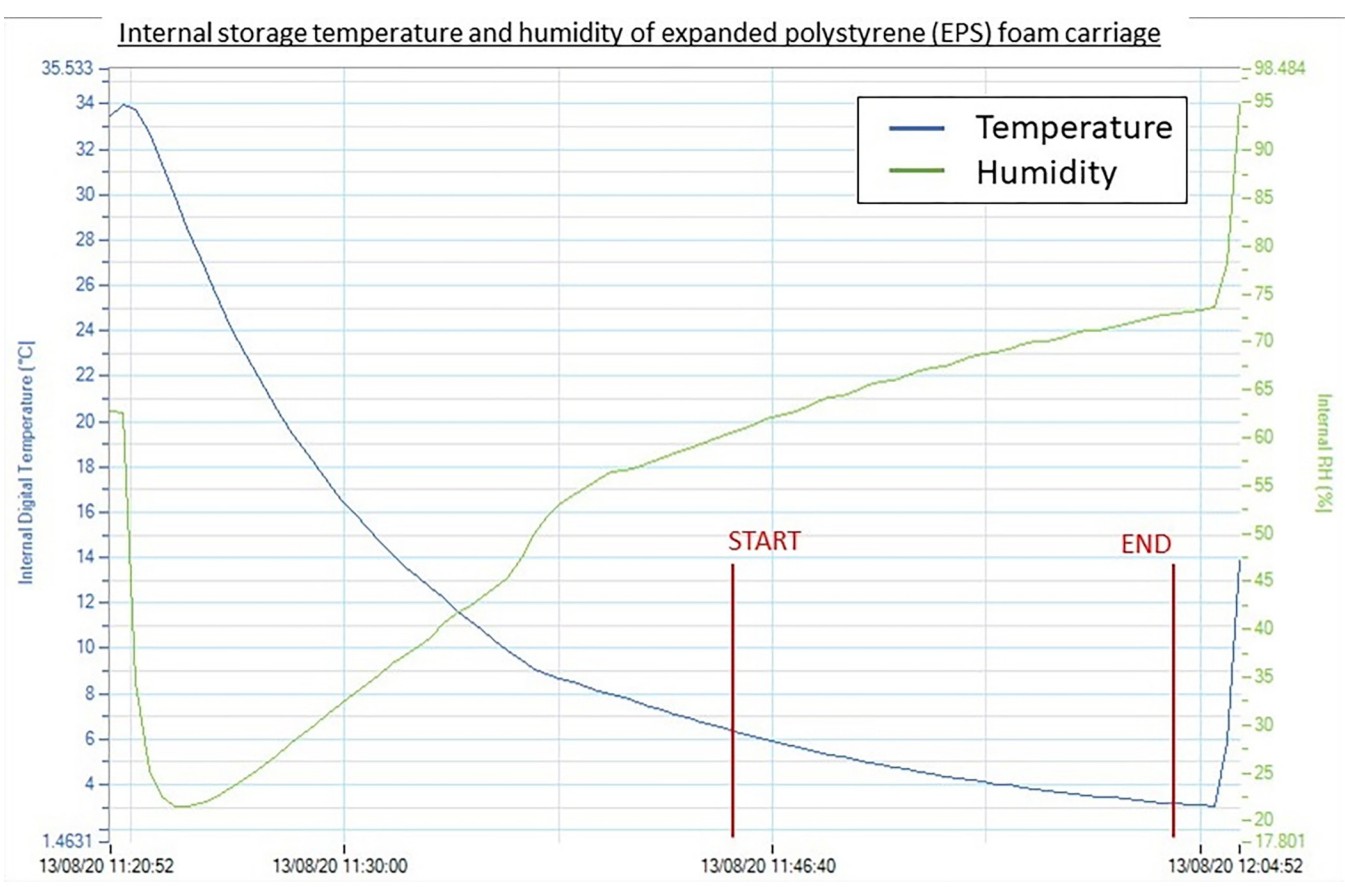

**Fig 4. Datalogger result of Flight 3.** Result in Flight 3 which used expanded polystyrene (EPS) foam drone carriage material showed the lowest and most stable result of mean kinetic temperature ±SD of 4.70 ± 1.14˚C.

Unalli et al. (2021) reported that the best storage temperature to maintain hematological analytes in EDTA tubes over 12 hours was 4˚C [30]. This result is in congruence with the current practice in Malaysia where the EPS foam storage box is used to transport blood samples using ground vehicles such as motorcycles or ambulance [31].

Phase 1 of our study was limited to observing temperature, overlooking other dimensions or details that may be influenced by the internal storage temperature of the drone carriage. To the best of our knowledge, no study has so far been conducted to compare the ability of

**Table 2. Mean and standard deviation (SD) of hematological and biochemical blood parameters from Phase 2 (F4 and F5) with the reference range.**

| Group | Blood sample parameters (mean ± SD) | | | | |
|---|---|---|---|---|---|
| | Hemoglobin (g/dL) | Hematocrit (%) | Hemolysis Index (mg/dL) | Potassium (mmol/L) | Sodium (mmol/L) |
| **Control (n = 10)** | 13.74 ± 1.97 | 41.18 ± 5.26 | 15.20 ± 13.39 | 5.7 ± 0.54 | 136.7 ± 1.34 |
| **Flight 4 (n = 10)** | 13.76 ± 2.01 | 41.05 ± 5.35 | 22.90 ± 24.23 | 5.7 ± 0.54 | 137.7 ± 1.37 |
| **Flight 5 (n = 10)** | 13.72 ± 2.00 | 41.31 ± 5.28 | 30.60 ± 39.56 | 5.8 ± 0.64 | 136.8 ± 1.03 |
| **Reference range***| 14.0–18.0 (Adult male) | 40–54 (Adult male) | 0–50 | 3.0–5.1 | 135–145 |
| | 12.0–16.0 (Adult female) | 37–47 (Adult female) | | | |

*Data source of reference ranges from Haematology and Chemical Pathology Units, Department of Diagnostic Laboratory Services, Universiti Kebangsaan Malaysia (UKM) Medical Centre, Kuala Lumpur, Malaysia.

**Table 3. Statistical analysis of Phase 2 results using Mann-Whitney U test.**

| Parameters | P-values | |
|:---:|:---:|:---:|
| | **F4 versus Control (n = 10)** | **F5 versus Control (n = 10)** |
| **Hemoglobin** | 0.970 | 0.970 |
| **Hematocrit** | 0.850 | 0.850 |
| **Hemolysis Index** | 0.570 | 0.623 |
| **Potassium** | 0.940 | 0.970 |
| **Sodium** | 0.095 | 0.785 |

different materials in maintaining storage temperature and the effects on the quality of blood samples during drone transportation. We did not conduct actual measurements of the properties of the three materials (conductivity, mechanical strength, etc) but relied on published values. We emphasized on this matter based on several previous studies which stated that the most important aspect to ensure good quality in blood samples transportation was temperature, followed by vibration regardless of the type of transport vehicle [32]. These results are crucial as a future reference in setting the path for the direction of such research on blood samples transportation using drones in tropical countries.

Yakushiji et al. (2021) reported that the proposed temperature for storage and transport of blood samples is slightly different according to various countries and climate such as the United Kingdom (2–10˚C) and the USA (1–10˚C) [33]. However, the recommendation by the International Society of Blood Transfusion is to adopt a transport temperature range of 2–6˚C [34]. Optimal temperature during transportation, regardless of the mode of transportation, will ensure blood samples quality and viability, maintain its biochemical properties and reduce blood wastage [35].

In Phase 2, we conducted two drone flights that transported blood samples in order to observe any influence of the hot equatorial climate on the quality of the samples. The results showed that the blood quality remained stable and was not significantly different from control samples. Our findings indicate that drone transportation with EPS foam as the drone carriage material is safe for blood samples delivery in equatorial climates such as in Malaysia.

Our observations in Phase 2 compare favourably with the outcome of a study conducted in Japan, where drone transportation of blood samples (referred to as red blood cell solutions in the paper) did not alter the level of lactate dehydrogenase, which was the blood parameter used as a hemolytic indicator in their study [36]. In the temperate Japanese climate and environment with cold winters and warm summers, the researchers concluded that the drone is a viable mode of blood samples transportation as it did not alter blood sample parameters.

In managing emergency cases in a remote location or during natural disasters such as landslides, mudslides, or floods, the ability of drones to transport blood samples over a distance of 8 km safely without compromising its quality, as in our study, can be life-saving as it assists clinical management [37]. Blood samples from remote areas for urgent blood tests can be analysed much faster with the use of drones to transport blood samples. Obstacles for ground vehicles that were previously described including geographical distances, challenging topographical features such as mountains and rivers, poor or underdeveloped road systems including untarred, uneven and narrow road conditions can be overcome with drones as a mode of transportation [38]. Therefore, the drone is a potential alternative to explore in tropical countries as a first response vehicle in emergencies.

Our research was limited to two flights carrying 60 blood samples. We were unable to increase the number of flights due to logistical constraints and the high cost of drone flights. Nonetheless, we believe that our findings on the blood sample parameters built sufficient

inroads for future research expansion. The observations from both drone flights were convincingly consistent. The usage of simulated blood to make up the drone payload as in our study has also been deployed in other drone studies elsewhere including in Montreal, Québec and Canada due to safety concerns [39].

## Conclusions

We conclude that, during transportation by drone, the internal storage temperature can be confidently maintained at optimum level and the hematological and biochemical integrity of blood samples remain stable and unperturbed when the drone carriage material used was EPS. The drone therefore appears to be a safe mode of transportation for blood samples in a hot tropical climate where it may serve as a solution to enhance healthcare accessibility, saving transport time during emergencies and providing a much wider healthcare coverage to the population. Future research is recommended to compare the effect of drone transportation on the biochemical quality of blood samples using EPS foam carriage at different geographical locations in various climates over longer flight distances and durations. Breakthrough of this advanced technology in the medical field will improve healthcare access in rural communities and resource-limited settings.

## Supporting information

**S1 File. TREND statement checklist.** The Transparent Reporting of Evaluations with Non-randomized Designs (TREND) statement checklist used for reporting this research.
(PDF)

## Acknowledgments

We would like to thank the Director General of Health Malaysia, Tan Sri Dr Noor Hisham Abdullah for his permission to publish this article (Reference Number: NIH.800-4/4/1. Jld 94 (30)). We thank our colleagues and staff from the Ministry of Health (MOH), Malaysia, who provided insight and expertise that greatly assisted this research. We gratefully thank our research team members from the Aerodyne Group Pte Ltd for their assistance and support throughout the study.

## Author Contributions

**Conceptualization:** Mohamed Afiq Hidayat Zailani, Ismail Mohd Saiboon, Shahnaz Irwani Sabri, Chan Kok Seong, Zaleha Abdullah Mahdy.

**Data curation:** Mohamed Afiq Hidayat Zailani, Raja Zahratul Azma Raja Sabudin, Aniza Ismail, Ismail Mohd Saiboon, Shahnaz Irwani Sabri, Zaleha Abdullah Mahdy.

**Formal analysis:** Mohamed Afiq Hidayat Zailani, Raja Zahratul Azma Raja Sabudin, Gan Kok Beng, Zaleha Abdullah Mahdy.

**Funding acquisition:** Zaleha Abdullah Mahdy.

**Investigation:** Jamaludin Mail.

**Methodology:** Mohamed Afiq Hidayat Zailani, Raja Zahratul Azma Raja Sabudin, Aniza Ismail, Rahana Abd Rahman, Ismail Mohd Saiboon, Jamaludin Mail, Gan Kok Beng, Zaleha Abdullah Mahdy.

**Project administration:** Ismail Mohd Saiboon, Shahnaz Irwani Sabri, Jamaludin Mail.

**Resources:** Shahnaz Irwani Sabri, Chan Kok Seong, Jamaludin Mail, Zaleha Abdullah Mahdy.

**Software:** Ismail Mohd Saiboon.

**Supervision:** Raja Zahratul Azma Raja Sabudin, Aniza Ismail, Rahana Abd Rahman, Zaleha Abdullah Mahdy.

**Validation:** Shamsuriani Md Jamal, Zaleha Abdullah Mahdy.

**Visualization:** Mohamed Afiq Hidayat Zailani, Chan Kok Seong.

**Writing – original draft:** Mohamed Afiq Hidayat Zailani.

**Writing – review & editing:** Raja Zahratul Azma Raja Sabudin, Aniza Ismail, Rahana Abd Rahman, Ismail Mohd Saiboon, Gan Kok Beng, Zaleha Abdullah Mahdy.

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
