## [Decision Letter · Decision Letter 0]

28 Mar 2022

PONE-D-22-02393Influence of drone carriage material on maintenance of storage temperature and quality of blood samples during transportation in an equatorial climatePLOS ONE

Dear Dr. Mahdy,

Thank you for submitting your manuscript to PLOS ONE. After careful consideration, we feel that it has merit but does not fully meet PLOS ONE’s publication criteria as it currently stands. Therefore, we invite you to submit a revised version of the manuscript that addresses the points raised during the review process.

We look forward to receiving your revised manuscript.

Kind regards,

Shiv Shankar

Academic Editor

PLOS ONE

Journal Requirements:

2. Please provide additional details regarding participant consent. In the ethics statement in the Methods and online submission information, please ensure that you have specified whether: 1) whether the ethics committee approved the verbal/oral consent procedure, 2) why written consent could not be obtained, and 3) how verbal/oral consent was recorded. If your study included minors, please state whether you obtained consent from parents or guardians in these cases. If the need for consent was waived by the ethics committee, please include this information.

3. We note you have included a table to which you do not refer in the text of your manuscript. Please ensure that you refer to Table 3 in your text; if accepted, production will need this reference to link the reader to the Table.

Reviewers' comments:

Reviewer's Responses to Questions

**Comments to the Author**

1. Is the manuscript technically sound, and do the data support the conclusions?

Reviewer #1: Partly

Reviewer #2: Yes

2. Has the statistical analysis been performed appropriately and rigorously? 

Reviewer #1: Yes

Reviewer #2: Yes

3. Have the authors made all data underlying the findings in their manuscript fully available?

Reviewer #1: Yes

Reviewer #2: Yes

4. Is the manuscript presented in an intelligible fashion and written in standard English?

Reviewer #1: Yes

Reviewer #2: Yes

5. Review Comments to the Author

Reviewer #1: The authors present an interesting paper, showing novel data that is of interest and within scope of PLOS.

However there are a number of points that must be addressed before publication.

1) Lines 46-47 please add statistics and analysis of the error to the temperature data given in the abstract.

2) Lines 73 to 74. please provide seasonal values for humidity and rainfall that matches the range of temperatures given in line 74.

3) Line 77. Please describe how blood samples are currently transported, I.e. describe the business as usual and also describe the packaging solutions / materials used for this current logistical transportation.

4) At the end of the introduction please add an explicit aim.

5) Within the study design please provide the required specification for the transport box / or carriage container. What are the required internal and external dimensions? What is the maximum and ideal weights for this box?

6) Lines 100 to 102 please provide a rationale for the selection of the different types of carriage materials.

7) Line 102 and or table 1: Please provide the grades, suppliers and thickness of the three materials.

8) Table 1: For all of the properties, e.g. Conductivity, mechanical strength, etc, please provide data and the required ranges in the specification and the actual values for the 3 samples used.

9) Table 1: Furthermore please consider adding the following. Modulus, yield strength, density, cost and environmental impact.

10) Table 1: Please cite references to support the parameters discussed.

11) Line 112: Please describe the payload capacity of the drone in question.

12) Line 114: The IUPAC agreed units are mL and not ml.

13) Line 139:What prior information was given to the donors?

14) Line 148: Were the flights BVLOS, if so how was the drone controlled and what were the safety measures put in place.

15) Line 157: Please provide more information about the simulated blood, what was the composition?

16) Line 191: Please describe the thickness of the materials and how the boxes were formed and held together.

17) Line 221: Please describe the shape and dimensions of the box. Also describe how the samples were packed within the box. Was UN3373 compatible packaging used?

18) Page 12: Why wasn't constant temperature rooms or incubators used to test the boxes before flight?

19) Please consider adding pictures of the materials used, the different boxes, and loading of samples. Also please add flight route maps etc.

20) Consider converting the temperature data into mean kinetic temperatures.

Reviewer #2: The article is well structured and deals with a relevant subject. However, I suggest that the authors complement the conclusions with the following information: suggestions for future research and research contribution - theoretical, practical, and social.

6. PLOS authors have the option to publish the peer review history of their article (what does this mean?). If published, this will include your full peer review and any attached files.

Reviewer #1: **Yes: **Dr Paul G. Royall

Reviewer #2: **Yes: **Claudia Araujo

---

## [Author Response · Author response to Decision Letter 0]

7 May 2022

Dear Editor,

We thank the reviewers for their invaluable comments. These are our point-by-point responses to the comments:

Comments of Reviewer 1

General Comment: The authors present an interesting paper, showing novel data that is of interest and within scope of PLOS. However, there are a number of points that must be addressed before publication.

Authors' Response: Thank you for your observation. We appreciate your suggestions for further improvement of this manuscript.

Comment 1: Lines 46-47 please add statistics and analysis of the error to the temperature data given in the abstract.

Authors' Response: We have added Standard Deviation (SD) values to the temperature data in the abstract as the analysis of error.

Lines 46-48: “In Phase 1, EPS foam was found to be the best material to maintain a stable and favorable internal storage temperature at a mean kinetic temperature ±SD of 4.70 ±1.14 °C. Much higher and unfavorable mean kinetic temperatures were recorded for aluminium (11.46 ±0.35 °C) and plastic (14.17 ±0.05 °C).

Comment 2: Lines 73 to 74. please provide seasonal values for humidity and rainfall that matches the range of temperatures given in line 74.

Authors' Response: We have provided the seasonal values of humidity and rainfall with references.

Lines 76-78: “Annually, average temperature across Malaysia is between 23°C and 34°C with average 80% annual rainfall (between 2000mm to 2500mm) and 80.5% annual percentage of humidity [17].”

Comment 3: Line 77. Please describe how blood samples are currently transported, I.e., describe the business as usual and also describe the packaging solutions / materials used for this current logistical transportation.

Authors' Response: We have added the description of current logistical transportation as follows:

Lines 80-85: “The current practice of logistical transportation of blood samples in Malaysia uses ground transportation such as motorcycles and ambulances. Once blood samples have been collected into tubes, they are stored inside a cooler box made of expanded polystyrene (EPS) foam material, with ice packs to maintain the internal storage temperature, and subsequently delivered to the nearest tertiary hospital.”

Comment 4: At the end of the introduction please add an explicit aim.

Authors' Response: We amended the paragraph and stated an explicit aim as follows:

Line 96: “The main objective of our study was to determine the most suitable material to be used as the drone carriage in order to maintain a favourable blood storage temperature within an equatorial climate, and its impact on the quality of blood samples carried by the drone.”

Comment 5: Within the study design please provide the required specification for the transport box / or carriage container. What are the required internal and external dimensions? What is the maximum and ideal weights for this box?

Authors' Response: We have added the details as follows:

Lines 117-124: “These drone carriages were within the optimum specification requirements for a reasonably sized and economically viable drone with a payload of 2.5 kilograms, that is planned for such a function in the long term. By excluding the weight of the contents inside the carriage such as blood samples, icepacks and thermologger totalling 1.5 kilograms, the ideal weight of the carriage therefore should not exceed 1 kilogram. In this study, the actual weights of the three carriages used were 0.2, 0.6 and 0.9 kilogram for PP plastic, EPS foam and aluminium, respectively. Their internal storage volume ranged from 4 to 6 liters, and their external dimensions (length x width x height) were 29 cm x 20 cm x 17 cm for PP plastic, 24 cm x 14 cm x 18 cm for EPS foam and 30 cm x 30 cm x 8 cm for aluminium.

Comment 6: Lines 100 to 102 please provide a rationale for the selection of the different types of carriage materials.

Authors' Response: We have added our rationale for the materials’ selection.

Line 114-116 : “The selected materials were chosen based on their capability to be made into a lightweight storage with good thermal insulation that is important to ensure safe drone transportation of blood samples. 

Comment 7: Line 102 and or table 1: Please provide the grades, suppliers and thickness of the three materials.

Authors' Response: We have added the requested details as follows:

Lines 110-112: “Three types of carriage materials were used: aluminium (Grade 6061 , 1 mm, Smartiff Ptd. Ltd., Malaysia), polypropylene (PP) plastic (Grade MMBX-8501, 3mm, Mysuppliersorg Pte. Ltd., Malaysia), and expanded polystyrene (EPS) foam (Grade NR-4153, 10mm, Mr. DIY Pte. Ltd., Malaysia), for flights F1, F2 and F3 respectively.”

Comment 8: Table 1: For all of the properties, e.g. Conductivity, mechanical strength, etc, please provide data and the required ranges in the specification and the actual values for the 3 samples used.

Authors' Response: We have added the data and required ranges as requested by the reviewer in Table 1 as follows:

Type of material

Aluminium

Polypropylene (PP) plastic

Expanded polystyrene (EPS) foam

Chemical composition

Silvery white metal element of Group 13 of the periodic table.

A transparent thermoplastic made from the combination of propylene monomers.

White foam plastic material produced from solid beads of polystyrene (hydrocarbon compound).

Density (kg/m3)

2680 – 2700 

910

12 – 46 

Elastic modulus

(Gpa)

70 – 80 

1.1-1.6

0.00650 - 2.65

Tensile strength 

(Mpa)

124 – 290 

27

0.8 – 1.1 

Yield strength

(Mpa)

195

35 – 40 

47 – 51

Thermal conductivity

[W/(m·K)]

151–202 

0.1 – 0.2 

0.035 – 0.037

Elongation at break

(%)

18 – 33 

50 – 145 

5 – 13 

Corrosion rate

(µm/year)

0.8 – 0.28 

< 0.1

< 0.1

Cost

(USD/kg)

0.93

1.42

0.98

Environmental impact

- Energy intensive (water, electricity, and resource)

- Greenhouse gas emissions

- Slow decomposition

(20 – 30 years )

- Toxic additive

-Easily recyclable

- Non-biodegradable

We did not measure actual values for the three samples used. We have added this aspect as a limitation of our study under the Discussion section: 

Lines 271-273: “We did not conduct actual measurements of the properties of the three materials (conductivity, mechanical strength, etc) but relied on published values.”

Comment 9: Table 1: Furthermore, please consider adding the following. Modulus, yield strength, density, cost and environmental impact.

Authors' Response: We have added the requested details in Table 1 (please refer to our response to comment No. 8).

Comment 10: Table 1: Please cite references to support the parameters discussed.

Authors' Response: We have added several in-text citations for the information displayed in Table 1.

Lines 116-117: “The physical characteristics of each material are summarized in Table 1 [20–24].”

Comment 11: Line 112: Please describe the payload capacity of the drone in question.

Authors' Response: We have added the following description:

Lines 133-139: “The number of samples was determined based on the weight of the samples versus the payload capacity of the drone. For this study, the drone model used had a payload capacity of 6.8 kilograms. After considering multiple external factors affecting the payload capacity and performance of the drone such as battery weight, drone weight, propeller number and size, and flight distance, we limited the total amount and weight of collected blood samples with their packaging and icepacks to suit our long term target of using a drone with a payload capacity of 2.5 kilograms [25].

Comment 12: Line 114: The IUPAC agreed units are mL and not ml.

Authors' Response: We have amended the unit from “ml” to “mL”.

Line 141: “All blood samples were drawn using a standard phlebotomy technique with a volume of 20 mL of blood from each subject.”

Comment 13: Line 139:What prior information was given to the donors?

Authors' Response: We have amended the “Patient and Public Involvement“ statement as follows:

Lines 173-176 : “Blood samples were taken from ten healthy donors following verbal consent. Information on the research were given to donors prior to obtaining their consent. It included the purpose and brief methodology of the research, the blood parameters that were to be measured, and the liberty of the donors to withdraw from the study at any time.”

Comment 14: Line 148: Were the flights BVLOS, if so how was the drone controlled and what were the safety measures put in place.

Authors' Response: The drone was flown within Visual Line of Sights (VLOS). We have added the description as follows:

Lines 181-183: The drone was flown within Visual Line of Sight (VLOS) with an average velocity of 42.9 km/h at an altitude of 300 feet above ground in a designated drone flying zone in Cyberjaya, Malaysia.”

Comment 15: Line 157: Please provide more information about the simulated blood, what was the composition?

Authors' Response: We have added the description as follows:

Lines 197-200: “We used simulated blood units consisting of distilled water that was dyed red (Star Brand Cochineal Red Artificial Food Colouring, Malaysia) instead of real blood packs in view of the highly valuable nature of real blood products and to eliminate the risk of a biohazard mishap in case of a drone crash or fall.”

Comment 16: Line 191: Please describe the thickness of the materials and how the boxes were formed and held together.

Authors' Response: We have amended and included the thickness of materials in Lines 110-112 (as stated in our response to comment no. 7)

We have added Figure 3 to illustrate how the carriages were formed and held together.

Lines 206-209: Figure 3 shows the attachment of the carriages to the drone.

 Fig 3. Attachment of carriages to drone. (a) Drone carriage made from EPS foam was attached to drone using a custom-made steel box with air cushion bags. (b) Drone carriage made from PP plastic was attached to drone using similar method as EPS foam. (c) Drone carriage made from aluminium was attached directly to drone using customized steel brackets.

Comment 17: Line 221: Please describe the shape and dimensions of the box.

Also describe how the samples were packed within the box. Was UN3373 compatible packaging used?

Authors' Response: We have added the following description:

Lines 259-261: “Our observation showed that EPS foam shaped into a six-faced cuboid with external dimensions (length x width x height) of 24 cm x 14 cm x 18 cm was the best material to be used for drone carriage…”

 Lines 144-150: “As a safety precaution, blood samples for the drone flights F4 and F5 were packed according to the UN3373 medical packaging regulations (Biological substance, Category B) [27]. This includes three compulsory components, namely the primary receptacles (the EDTA blood tubes) that were encapsulated and protected inside a secondary packaging of Low Density Polyethylene (LDPE) biohazard specimen bags, and lastly an outer packaging for transport (the drone carriage).”

Comment 18: Page 12: Why wasn't constant temperature rooms or incubators used to test the boxes before flight?

Authors' Response: We highly appreciate the reviewer’s insightful comment.

However, as we were able to fly the drone outdoors with blood samples, we did not use constant temperature rooms or incubators to test the boxes before flight because we preferred to test these in the real flight environment and climate in order to assess the maintenance of quality of blood samples for clinical use.

Comment 19: Please consider adding pictures of the materials used, the different boxes, and loading of samples. Also please add flight route maps etc.

Authors' Response: We have added the pictures as suggested, as Figure 2 (below) and Figure 3 (refer to response No. 16):

Lines 191-192: 

Fig 2. Drone flight path. Yellow lines illustrate the drone flight path used for all flights in this research.

Comment 20: Consider converting the temperature data into mean kinetic temperatures.

Authors' Response: We have calculated and converted all temperature data into mean kinetic temperatures. 

Lines 46-48: “…at mean kinetic temperature ±SD of…”

Line 228: “…the mean kinetic temperature ± SD of the…”

Line 237: “…stable result of mean kinetic temperature ±SD of…”

Line 262:”… to maintain optimal mean kinetic temperature for blood sample transportation…”

Comments of Reviewer 2

The article is well structured and deals with a relevant subject. However, I suggest that the authors complement the conclusions with the following information: suggestions for future research and research contribution - theoretical, practical, and social.

Authors' Response: Thank you for the good suggestion. We have added the suggestion in the Conclusion section as follows:

 Lines 321-325: “Future research is recommended to compare the effect of drone transportation on the biochemical quality of blood samples using EPS foam carriage at different geographical locations in various climates over longer flight distances and durations. Breakthrough of this advanced technology in the medical field will improve healthcare access in rural communities and resource-limited settings.”

---

## [Decision Letter · Decision Letter 1]

30 May 2022

Influence of drone carriage material on maintenance of storage temperature and quality of blood samples during transportation in an equatorial climate

PONE-D-22-02393R1

Dear Dr. Mahdy,

We’re pleased to inform you that your manuscript has been judged scientifically suitable for publication and will be formally accepted for publication once it meets all outstanding technical requirements.

Kind regards,

Shiv Shankar

Academic Editor

PLOS ONE

Additional Editor Comments (optional):

Reviewers' comments:

Reviewer's Responses to Questions

**Comments to the Author**

1. If the authors have adequately addressed your comments raised in a previous round of review and you feel that this manuscript is now acceptable for publication, you may indicate that here to bypass the “Comments to the Author” section, enter your conflict of interest statement in the “Confidential to Editor” section, and submit your "Accept" recommendation.

Reviewer #1: All comments have been addressed

Reviewer #2: All comments have been addressed

2. Is the manuscript technically sound, and do the data support the conclusions?

Reviewer #1: Yes

Reviewer #2: Yes

3. Has the statistical analysis been performed appropriately and rigorously? 

Reviewer #1: Yes

Reviewer #2: Yes

4. Have the authors made all data underlying the findings in their manuscript fully available?

Reviewer #1: Yes

Reviewer #2: Yes

5. Is the manuscript presented in an intelligible fashion and written in standard English?

Reviewer #1: Yes

Reviewer #2: Yes

6. Review Comments to the Author

Reviewer #1: Good morning Authors,

Many thanks for addressing all of the comments so well.

Really enjoyed re-reading the paper.

In the editing stage perhaps consider adding a scale bar to figure 2.

Reviewer #2: The authors improved the manuscript as recommended by the reviewers and the article is ready to be published. The new version is adequate and no further modifications are required.

7. PLOS authors have the option to publish the peer review history of their article (what does this mean?). If published, this will include your full peer review and any attached files.

Reviewer #1: **Yes: **Dr Paul G Royall

Reviewer #2: **Yes: **Claudia Araujo

---

## [Editor Report · Acceptance letter]

12 Aug 2022

PONE-D-22-02393R1 

Influence of drone carriage material on maintenance of storage temperature and quality of blood samples during transportation in an equatorial climate 

Dear Dr. Mahdy:

I'm pleased to inform you that your manuscript has been deemed suitable for publication in PLOS ONE. Congratulations! Your manuscript is now with our production department. 

Kind regards, 

on behalf of

Dr. Shiv Shankar 

Academic Editor

PLOS ONE